# Effects of Diet Supplemented with Excess Pyrroloquinoline Quinone Disodium on Growth Performance, Blood Parameters and Redox Status in Weaned Pigs

**DOI:** 10.3390/ani11020359

**Published:** 2021-02-01

**Authors:** Dongxu Ming, Caiyun Huang, Wenhui Wang, Zijie Wang, Chenyu Shi, Xindi Yin, Linlin Sun, Youjun Gao, Fenglai Wang

**Affiliations:** 1State Key Laboratory of Animal Nutrition, College of Animal Science and Technology, China Agricultural University, Beijing 100193, China; mdx920825@163.com (D.M.); huangcaiyun@cau.edu.cn (C.H.); wangwh1025@cau.edu.cn (W.W.); wzjgf191114@163.com (Z.W.); scyshichenyu@163.com (C.S.); xyin01@uoguelph.ca (X.Y.); 2Shanghai Medical Life Science Research Center Company, Shanghai 200032, China; nt_shirley@163.com; 3Changmao Biochemical Engineering Company, Changzhou 213000, China; gyjandjs@163.com

**Keywords:** pyrroloquinoline quinone, safety, redox status, weaned pigs

## Abstract

**Simple Summary:**

Weaning is a vital process for weaned pigs since piglets are exposed to psychologic and environmental stresses. These stresses converge on the pig to cause low feed consumption and weight gain meanwhile increased risk of diarrhea and mortality during the early postweaning period. The use of antibiotic growth promoters to help prevent weaning stress in weaned pigs has been forbidden in the European Union, Korea, Japan and China. Pyrroloquinoline quinone disodium (PQQ·Na_2_) is increasing interest in use of alternatives to in-feed antibiotics. In this study, we found PQQ·Na_2_ can improve growth performance meanwhile improves antioxidant status of weaned pigs. A high oral dose of PQQ·Na_2_ does not appear to have harmful effects on weaned pigs.

**Abstract:**

The research was implemented to assess the safety of feeding excess of pyrroloquinoline quinone disodium (PQQ·Na_2_) to 108 Duroc × Landrace × Large White weaned pigs (BW = 8.38 ± 0.47 kg). Pigs were weaned at 28 d and randomly distributed to one of three diets with six replicates and six pigs per replicate (three males and three females). Pigs in the control group were fed a corn-soybean meal-based diet (without growth promoter) while the two experimental diets were supplied with 7.5 and 75.0 mg/kg PQQ·Na_2_, respectively. Average daily gain (ADG), average daily feed intake (ADFI), feed conversion (F:G), diarrhea incidence, hematology, serum biochemistry, organ index and general health were determined. Diets supplementation with 7.5 mg/kg PQQ·Na_2_ in weaned pigs could increase ADG during the entire experimental period (*p* < 0.05). And there was a tendency to decrease F:G (*p* = 0.063). The F:G of weaned pigs fed 7.5 and 75.0 mg/kg PQQ·Na_2_ supplemented diets was decreased by 9.83% and 8.67%, respectively, compared to the control group. Moreover, pigs had reduced diarrhea incidence (*p* < 0.01) when supplemented with PQQ·Na_2_. No differences were observed between pigs supplemented with 0.0, 7.5 and 75.0 mg/kg PQQ·Na_2_ diets on hematological and serum biochemical parameters as well as histological assessment of heart, liver, spleen, lung and kidney. At day 14, pigs had increased activity of glutathione peroxidase (GSH-Px) (*p* < 0.05), catalase (CAT) (*p* < 0.05) and total antioxidant capacity (T-AOC) (*p* < 0.05), and the serum concentration of malondialdehyde (MDA) was decreased (*p* < 0.01) with PQQ·Na_2_ supplementation. At day 28, pigs had increased activities of total superoxide dismutase (T-SOD) (*p* < 0.01), GSH-Px (*p* < 0.01), CAT (*p* < 0.05) and T-AOC (*p* < 0.01), and serum concentration of MDA was lower (*p* < 0.01) with PQQ·Na_2_ supplementation. In conclusion, PQQ·Na_2_ can improve weaned pigs growth performance and serum antioxidant status. Meanwhile high PQQ·Na_2_ inclusion of 75.0 mg/kg does not appear to result in harmful effects on growth performance of pigs.

## 1. Introduction

Pyrroloquinoline quinone (PQQ) was initially isolated from a microorganism and is a quinone compound like a coenzyme for methanol dehydrogenase [1,2,3]. PQQ has been found in both plant and animal tissues [4], and can be reversibly transformed into pyrroloquinoline quinol (PQQH_2_) via a semiquinone intermediate [5]. The ability of PQQH_2_ to eliminate free radicals is 7.4-fold higher than vitamin C [6], thus PQQ can reduce oxidative stress in organisms. PQQ plays an important mission in a lot of physiological and biological functions [7]. In particular, its role in growth promoting, anti-diabetic effects, anti-oxidative action and neuroprotection is widely concerned [8]. PQQ in mouse and rat models, can improve performance of growth, enhance reproductive and immunity, protect nerves and preserve myocardia function [9,10,11]. Supplementation with 0.2 mg/kg PQQ·Na_2_ for broiler chicks improved growth performance, dressing percentage, and immunity and plasma status [12,13]. In addition, diet supplementation with 0.1 mg/kg PQQ·Na_2_ can regulate meat quality and antioxidant ability in broilers [14]. 

The safety of PQQ has been extensively researched. In rats, the no-observed-adverse-effect-level (NOAEL) of PQQ·Na_2_ is 400 mg /kg bw/day for both sexes [15]. The median lethal dose of PQQ in rats was shown to be 500–1000 mg/kg body weight in female and 1000–2000 mg/kg body weight in male [16]. A research in human lymphocytes and mice proved that no genotoxic of PQQ was observed in vivo or in vitro [17]. In the United States, a maximum dose of PQQ for healthy adults is 50 mg/day [16].

Our previous study shown dietary supplementation PQQ·Na_2_ improved weaned pigs growth performance and decreased incidence of diarrhea by improving intestinal development [18]. And PQQ·Na_2_ can enhance weaned pigs’ immunity by inhibiting the NF-κB pathway meanwhile correcting intestinal microbiota maladjustment [19]. A study shown that PQQ·Na_2_ can protect weaned pigs’ intestines via modulating NQO1 related genes [20]. Gestating and lactating sows’ diets supplementation with 20 mg/kg PQQ·Na_2_ can improve their reproductive performance [21].

In pigs, little is known about the effects of PQQ overdoses on their growth and health. Therefore, our objective was to determine the influence of PQQ·Na_2_ on growth, blood parameters, organ indexes and redox status of weaned pigs. And the hypothesis is that excess PQQ·Na_2_ will not appear to have harmful effects on weaned pigs.

## 2. Materials and Methods

The protocol employed in this trial was approved by the China Agricultural University Animal Care and Use Committee (Beijing, China). (China Agricultural University Laboratory Animal Welfare and Animal Experimental Ethical Inspection Form No. AW3004020.)

### 2.1. Pigs and Experimental Protocol

A total of 108 Duroc × Landrace × Large White weaned pigs (28 ± 2 days of age; BW = 8.38 ± 0.47 kg), were allocated to one of three diets in a randomized complete block design with six replicates and six pigs per replicate (three barrows and three gilts). Pigs in the control group were fed a corn-soybean meal-based diet (without growth promoter) while pigs assigned to the two experimental treatment received the control diet supplemented with 7.5 or 75.0 mg/kg PQQ·Na_2_, respectively (Table 1). The corn-soybean meal-based diet was formulated to exceed National Research Council (2012) recommendations for weaned pigs [22]. The PQQ·Na_2_ (purity, ≥ 98%) was chemically synthesized and purchased from Changmao Biochemical Engineering Company (Changzhou, China). PQQ·Na_2_ was premixed with corn starch to a concentration of 1 g/kg mixture before inclusion in the diet. 

Pigs were housed in totally slatted floor pens (1.2 × 2.0 m^2^) containing a nipple drinker and stainless-steel feeder for ad libitum access to water and feed over the 28d trial. Ambient room temperature was maintained at 29 °C for the first week, and lowered by 1 °C each week thereafter. Diarrhea scores were recorded daily for all pigs from d 1 to 28 by the same person and were based on the following scale: 1 = well-formed feces, 2 = sloppy feces, 3 = diarrhea [23]. Pigs with a score of 3 were considered to have diarrhea. The incidence of diarrhea for weaned pigs in each pen was calculated as [(number of weaned pigs with diarrhea × number of days of diarrhea) / (total number of weaned pigs × number of days of experiment)] × 100 [24]. Body weight were measured on day 0, 14 and 28 and feed disappearance on these days were recorded. ADG, ADFI, F:G were calculated based on weight and feed measurements.

### 2.2. Sample Collection and Processing

On day 14 and d 28, after an overnight fast, 18 pigs (1 pig per pen) were selected for blood sampling according to body weight closest to the mean value within the pen. Blood samples were collected from the jugular vein into a vacuum tube containing EDTAK_2_ (ethylenediaminetetraacetic acid disodium salt) and no anticoagulant for whole blood and serum, respectively. Whole blood was assayed for hematological parameters within 1 h after sampling. Serum were separated by centrifugation for 10 min at 3000× *g* and 4 °C, and stored at −20°C until analysis.

On day 28, 18 weaned pigs (1 pig per pen) were weighed, electrically stunned and immediately euthanized by exsanguination. Heart, liver, spleen, lung and kidney were harvested and weighed. Relative weights of heart, liver, spleen, lung and kidney were expressed as a ratio to whole body mass (g/kg). After weighed, immediately take tissue samples fixed in 4% formalin buffer solution.

### 2.3. Hematological and Serum Biochemical Parameters Analysis

Hematological parameters including red blood cells (RBC), white blood cells (WBC), hematocrit (HCT), hemoglobin (HGB), mean corpuscular hemoglobin (MCH), mean corpuscular volume (MCV), mean corpuscular hemoglobin concentration (MCHC), platelet count (PLT) and red blood cell distribution width-coefficient of variance (RDW-CV) were determined using a Sysmex Microcell Counter CL-180 (Tokyo, Japan). 

Serum biochemical parameters including glucose (GLU), aspartate aminotransferase (AST), alanine aminotransferase (ALT), alkaline phosphatase (ALP), albumin (ALB), total bilirubin (TBILI), urea nitrogen (UN), total protein (TP) and creatinine (CRE) were measured by corresponding commercial kits (BioSino Bio-technology and Science Incorporated, Beijing, China) using an Automatic Biochemical Analyzer (Hitachi 7160, Hitachi High Technologies Corporation, Tokyo, Japan).

### 2.4. Histopathology Analysis

Tissue samples from the heart, liver, spleen, lung and kidney were fixed in 4% formalin buffer solution for 48 h, cleaning and embedded in paraffin. Paraffin sections of 5μm thickness were stained with hematoxylin and eosin.

### 2.5. Antioxidative Physiological Analyses

The T-SOD, GSH-Px, CAT, T-AOC and MDA activities in serum were determined using assay kits in strictly according to the manufacturer instructions (Nanjing Jiancheng Bioengineering Institute, Nanjing, China).

### 2.6. Statistical Analysis

All experimental data except for diarrhea incidence were analyzed using the GLM Procedure of SAS as a randomized complete block design (SAS Inst. Inc., Cary, NC, USA). Differences in diarrhea incidence among treatments were tested by a Chi-square test. Pen was the experimental unit for the performance traits, and individual pig was the experimental unit for all other traits studied. Differences among means were evaluated by the Student-Newman-Keuls test. And all data except the diarrhea incidence of the experiment followed a normal distribution. Treatment effects were considered to be significant if *p* < 0.05.

## 3. Results

### 3.1. Performance and Diarrhea Incidence

The effects of PQQ·Na_2_ supplementation on ADG, ADFI and F:G are presented in Table 2. During the entire experimental period, weaned pigs fed 7.5 mg/kg PQQ·Na_2_ supplemented diets displayed increased ADG (*p* < 0.05) compared to control. No difference was noted in ADFI among the different groups (*p* > 0.05). And there was a tendency to decrease F:G (*p* = 0.063). Overall, weaned pigs fed 7.5 and 75.0 mg/kg PQQ·Na_2_ supplemented diets was decreased by 9.83% and 8.67%, respectively, compared to the control group but these differences were not statistically significant.

Diarrhea incidence was exhibited in Table 3, weaned pigs fed diets supplemented with 7.5 or 75.0 mg/kg PQQ·Na_2_ had decreased diarrhea incidence (*p* < 0.01) than weaned pigs fed diets non-supplemented with PQQ·Na_2_ during days 0 to 14, days 15 to 28 and the entire experimental period.

### 3.2. Hematological and Serum Biochemical

All hematologic and serum biochemical parameters were present in Table 4 and Table 5. Normal hematologic and serum biochemical parameters reference values is from Merck Veterinary Manual (Merck, 2010). There were no significant differences observed in hematologic and serum biochemical parameters that weaned pigs supplemented with 0.0, 7.5 or 75.0 mg/kg PQQ·Na_2_. 

### 3.3. Organ Index and Histological Structures

Effects of graded levels of PQQ·Na_2_ on organ index in weaned pigs are presented in Table 6. Histological structures of heart, liver, spleen, lung and kidney were observed in pigs fed different graded levels of PQQ·Na_2_ diets for 28 days are presented in Figure 1.

When weaned pigs fed diets supplemented with 0.0 (control), 7.5 (low-dose) or 75.0 (high-dose) mg/kg PQQ·Na_2_, normal histological structures in the heart, liver, spleen and kidney were observed. The lungs of all experimental animals showed interstitial pneumonia characterized by massive infiltration of lymphoid cells.

### 3.4. Redox Status

Data for antioxidative enzymes in serum are presented in Table 7. At day 14, pigs fed diets supplemented with 7.5 mg/kg PQQ·Na_2_ had increased activity of CAT (*p* < 0.05) and T-AOC (*p* < 0.05), and the serum concentration of MDA (*p* < 0.01) was decreased compared with control. And pigs fed diets supplemented with 75.0 mg/kg PQQ·Na_2_ had increased activity of GSH-Px (*p* < 0.05) and T-AOC (*p* < 0.05) compared with control. At d 28, pigs fed diets supplemented with 7.5 mg/kg PQQ·Na_2_ had increased activities of T-SOD (*p* < 0.01), GSH-Px (*p* < 0.01) and CAT (*p* < 0.05), and serum concentration of MDA (*p* < 0.01) was lower compared with pigs fed no PQQ·Na_2_. Pigs fed diets supplemented with 75.0 mg/kg PQQ·Na_2_ had increased activities of T-SOD (*p* < 0.01), GSH-Px (*p* < 0.01) and T-AOC (*p* < 0.01), and serum concentration of MDA (*p* < 0.01) was lower compared with control.

## 4. Discussion

Weaning is a necessary step in pig production. During weaning, piglets are subjected to a variety of psychological and environmental stresses. Among these stressors, the weaning diet is plant-based, less digestible and contains antinutritional factors. Furthermore, the pig is accustomed to a liquid diet before weaning and is abruptly changed to a solid diet after weaning. These stresses converge on the pig to cause low feed consumption and weight gain meanwhile increased risk of diarrhea and mortality during the early postweaning period. The use of antibiotic growth promoters to help prevent weaning stress in weaned pigs has been forbidden in the European Union, Korea, and Japan [25]. And in China the use of antibiotic growth promoters also has been forbidden in 2020 [26]. Consequently, here is increasing interest in use of alternatives to in-feed antibiotics. It is reported that, PQQ·Na_2_ improved growth performance and decreased diarrhea by enhancing the intestinal morphology and improving the redox status in weaned pigs [18]. And whether PQQ·Na_2_ can be used as a substitute for in-feed antibiotics, we need to consider adding the positive control (with standard animal growth promoter added) in our future research.

In 2009, PQQ·Na_2_ was commercialized in the United States after approval through the Food and Drug Administration. The safety of PQQ has been wide researched. A research found that the oral median lethal dose of PQQ·Na_2_ for male and female rats are 1000–2000 mg/kg body weight and 500–1000 mg/kg body weight, respectively. And the NOAEL of PQQ·Na_2_ is 100 mg /kg bw/d for both sexes [16]. Thereafter, another research indicates that the NOAEL of PQQ·Na_2_ is 400 mg /kg bw/d for both sexual rats [15]. The neuroprotective effect of PQQ has also attracted wide attention. In order to treat human psychological and neurological disorders with PQQ, research found that the NOAEL of PQQ intracephalic injection to mice cortical neurons is 1 μM [27]. A research demonstrated that PQQ·Na_2_ does not have genotoxic potential in vivo [17]. Other studies report that the pharmacokinetic behavior of PQQ Na_2_ is similar to other water-soluble B vitamin compounds. This suggests that PQQ will not accumulate in the body to produce serious damage [28]. However, intraperitoneal injection of PQQ Na_2_ at a dose of 11.5 mg/kg body weight for 4 consecutive days, morphologic and functional changes of the kidney were observed [29]. Future more, PQQ can against CTX-induced nephrotoxicity [30]. From these studies, oral administration of PQQ appears to have minimal negative effects, whereas it is necessary to monitor for super dosage.

The hematologic and serum biochemical profiles correlated with pigs’ health. And supplementation with 7.5 and 75.0 mg/kg PQQ·Na_2_ have no significant differences compared with control group in hematologic and serum biochemical parameters. Histological structures in the lung have a certain proportion of lesion when pigs fed the different levels of PQQ·Na_2_ for 28 days. Probably during the experiment, pigs were generally infected with pathogenic microorganisms, the proportion in control group, low-dose group and high-dose group were similar. And supplementation with 7.5 and 75.0 mg/kg PQQ·Na_2_ can obviously improve the structure of liver and promote the development of splenic white medulla lymphatic follicles. Our results obtained from the current research reflected that weaned pigs take excess PQQ·Na_2_ will not lead to harmful influence to health.

A study proved that weaning will cause oxidative stress injury which plays an important role in free radical metabolism [31]. Free radicals are molecules produced during aging, tissue damage and oxidative stress, if not removed by antioxidants, they may be toxic to cellular components which could induce diarrhea [32,33]. Superoxide dismutase (SOD) provides competent dismutation of O^2−^ and converts it to hydrogen peroxide, which is removed by catalase and GSH-Px [34]. Therefore, increases in GSH-Px, CAT and SOD enzyme levels may be suggesting alleviation of oxidative stress. Some researchers have proved that reduced forms of PQQ (PQQH_2_) have obvious antioxidant ability [35,36,37,38]. The ability of PQQH_2_ to eliminate free-radicals was 7.4-fold higher than vitamin C, which is known as the most active water-soluble antioxidant [6]. The T-AOC is an oxidative stress and antioxidant defense biomarker that forms the first step in search for a healthy organism status [39]. In this experiment, we examined the levels of T-SOD, GSH-Px, CAT and T-AOC in serum, and determined that PQQ·Na_2_ supplementation increased activity of antioxidant enzymes and T-AOC, which suggests alleviation of oxidant stress may have occurred. MDA is a naturally occurring product of lipid peroxidation and prostaglandin biosynthesis [40] and is an indicator of lipid peroxidation. Supplementation of PQQ·Na_2_ reduced serum concentration of MDA which suggests lower levels of oxidative stress in PQQ·Na_2_-fed pigs.

In the present study, whether higher dose of PQQ·Na_2_ supplementation in diet has a positive effect on piglets’ growth performance and health state has been researched. We found diets supplemented 7.5 mg/kg PQQ·Na_2_ in weaned pigs can improve ADG compared to control. And compared to the control group, supplemented with 7.5 mg/kg PQQ·Na_2_ and a dosage 10 times higher (75.0 mg/kg PQQ·Na_2_) has a tendency to decrease F:G, but no impact on ADFI. However, in this research, we found diets supplemented with 7.5 and 75.0 mg/kg PQQ·Na_2_ in weaned pigs has no significant difference in weaned pigs growth performance. Similarly, the redox status of weaned pigs has no significant difference when diets supplemented with 7.5 and 75.0 mg/kg PQQ·Na_2_. Meanwhile, weaned pigs hematological and serum biochemical parameters, organ index and histological structures has no differences among the three treatment.

## 5. Conclusions

In conclusion, diets supplemented with 7.5 and 75.0 mg/kg PQQ·Na_2_ can improve growth performance meanwhile improves antioxidant status of weaned pigs. And there are no differences diets between supplemented with 7.5 and 75.0 mg/kg PQQ·Na_2_. A high oral dose of PQQ·Na_2_ (75.0 mg/kg PQQ·Na_2_) does not appear to have harmful effects on weaned pigs.

## Figures and Tables

**Figure 1 animals-11-00359-f001:**
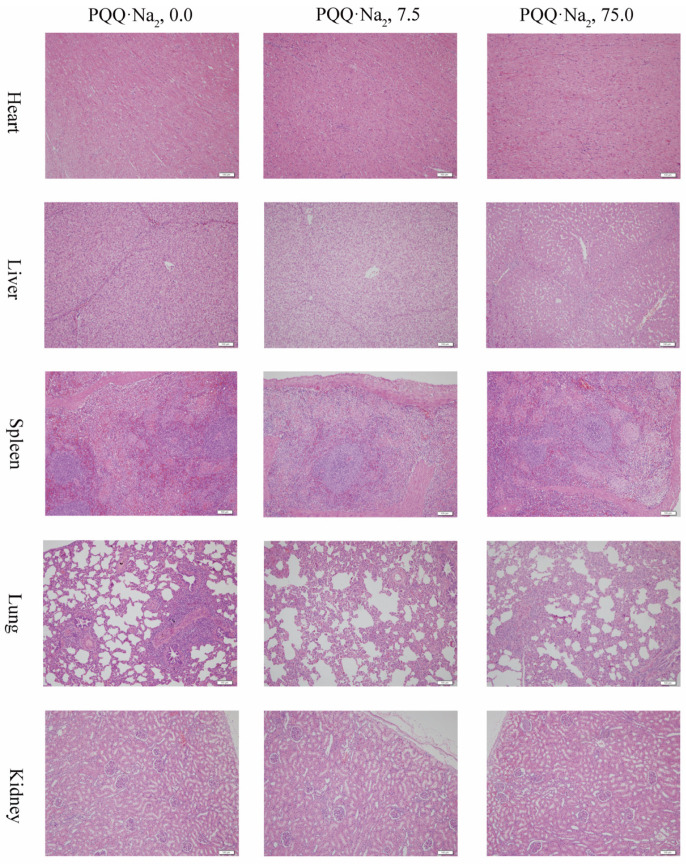
Effects of graded levels of PQQ·Na_2_ on heart, liver, spleen, lung and kidney morphology in weaned piglets. Hematoxylin-eosin staining, scale bars = 100 μm.

**Table 1 animals-11-00359-t001:** Composition and analyzed nutrient composition of experimental diets (as-fed basis).

Item	PQQ·Na_2_ ^1^_,_ mg/kg Diet		
	0	7.5	75.0
Ingredients, %			
Corn	59.34	58.59	51.84
Soybean meal	15.00	15.00	15.00
Extruded soybean	5.00	5.00	5.00
Soy protein concentrate	2.00	2.00	2.00
Fish meal	3.00	3.00	3.00
Dried whey	6.00	6.00	6.00
SDPP ^2^	2.00	2.00	2.00
Soybean oil	2.20	2.20	2.20
Sucrose	2.00	2.00	2.00
Limestone	0.90	0.90	0.90
Dicalcium phosphate	1.00	1.00	1.00
Salt	0.25	0.25	0.25
L-Lysine·HCl	0.48	0.48	0.48
L-Threonine	0.16	0.16	0.16
L-Tryptophan	0.05	0.05	0.05
L-Methionine	0.12	0.12	0.12
PQQ·Na_2_ premix ^3^	0.00	0.75	7.50
Vitamin and mineral premix ^4^	0.50	0.50	0.50
Nutrient levels ^5^, %			
Digestible energy, MJ/kg	15.10	14.98	14.02
Crude protein	18.52	18.45	18.50
Lysine	1.50	1.50	1.50
Methionine	0.40	0.41	0.40
Threonine	0.84	0.84	0.84
Tryptophan	0.25	0.24	0.24
Calcium	0.81	0.79	0.81
Total phosphorus	0.65	0.64	0.64

^1^ PQQ·Na_2_: pyrroloquinoline quinone disodium. ^2^ SDPP: spray-dried porcine plasma. ^3^ PQQ·Na_2_ was diluted with corn starch to a concentration of 1.0 g/kg mixture. ^4^ Provided the following vitamins and minerals per kg of diet: vitamin A, 11,000 IU as retinyl acetate; vitamin D_3_, 1500 IU as cholecalciferol; vitamin E, 44.1 IU as DL-α-tocopherol acetate; vitamin K_3_, 4 mg as menadione; vitamin B_1_, 1.4 mg; vitamin B_2_, 5.2 mg; vitamin B_5_, 20 mg; vitamin B_12_, 10 μg; niacin, 26 mg; pantothenic acid, 14 mg; folic acid, 0.8 mg; biotin, 44 μg; Fe, 100 mg from FeSO_4_; Cu, 16.5 mg from CuSO_4_·5H_2_O; Zn, 90 mg from ZnO; Mn, 35 mg from MnSO_4_; I, 0.3 mg from KI; Se, 0.3 mg from Na_2_SeO_3_. ^5^ Digestible energy are calculated values. Other nutrient levels in the table are analyzed values.

**Table 2 animals-11-00359-t002:** Performance of weaned pigs fed graded levels of PQQ·Na_2_
^1^ for 28 day ^2.^

Item	PQQ·Na_2_, mg/kg Diet			SEM	*p*-Value
	0	7.5	75.0		ANOVA
Body weight					
0 day, kg	8.38	8.45	8.32	0.47	0.983
14 day, kg	13.87	14.29	13.84	0.67	0.883
28 day, kg	19.95	21.50	20.61	0.78	0.396
days 0 to 14 ^3^					
ADG, g/day	423	477	444	21	0.197
ADFI, g/day	628	664	611	40	0.664
F:G	1.48	1.39	1.37	0.06	0.391
days 15 to 28					
ADG, g/day	434	492	473	17	0.054
ADFI, g/day	848	848	822	38	0.876
F:G	1.96	1.72	1.74	0.09	0.107
days 0 to 28					
ADG, g/day	429 ^b^	487 ^a^	455 ^ab^	16	0.035
ADFI, g/day	742	759	720	33	0.732
F:G	1.73	1.56	1.58	0.06	0.063

^1^ PQQ·Na_2_: pyrroloquinoline quinone disodium. ^2^ Values are means of 6 pens (6 pigs/pen, experiment unit was the pen) and pooled SEM (standard error of the mean), *n* = 6. Means within a row without common superscripts differ (*p* < 0.05). ^3^ADG: average daily gain; ADFI: average daily intake; F:G: feed conversion (feed: gain).

**Table 3 animals-11-00359-t003:** Incidence of diarrhea in weaned pigs fed graded levels of PQQ·Na_2_
^1^ for 28 day ^2.^

Item	PQQ·Na_2_, mg/kg Diet			SEM	*p*-Value
	0	7.5	75.0		
Diarrhea incidence,^3^ %					
Days 0 to 14	18.22 ^a^	13.35 ^b^	12.11 ^b^	0.75	<0.01
Days 15 to 28	11.86 ^a^	6.46 ^b^	6.74 ^b^	0.54	<0.01
Days 0 to 28	15.10 ^a^	10.02 ^b^	9.85 ^b^	0.49	<0.01

^1^ PQQ·Na_2_ = pyrroloquinoline quinone disodium. ^2^ Values are means of 6 pens (6 pigs/pen, experiment unit was the pen) and pooled SEM (standard error of the mean), *n* = 6. Means within a row without common superscripts differ (*p* < 0.05). ^3^ Number of pigs with diarrhea in each pen × diarrhea days/ (total number of pigs × 28 d) × 100.

**Table 4 animals-11-00359-t004:** Effects of graded levels of PQQ·Na_2_
^1^ on hematological parameters of weaned piglets ^2.^

Item ^3^	Reference Value ^4^	PQQ·Na_2_, mg/kg Diet			SEM	*p*-Value
		0	7.5	75.0		ANOVA
day 14						
WBC, 10^9^/L	11–22	19.92	17.85	18.27	1.10	0.393
RBC, 10^12^/L	5–7	6.01	6.26	6.02	0.20	0.692
HGB, g/L	90–130	101.67	110.33	103.27	3.31	0.140
HCT, %	36–43	35.82	38.13	35.75	1.04	0.189
MCV, fL	52–62	59.33	60.88	59.60	1.15	0.621
MCH, pg	17–24	16.82	17.63	17.18	0.35	0.279
MCHC, g/L	290–340	283.83	289.17	288.67	2.28	0.195
PLT, 10^9^/L	200–500	459.00	408.83	504.17	68.97	0.648
RDW-CV, %	-	18.77	18.08	18.55	0.66	0.778
day 28						
WBC, 10^9^/L	11–22	19.13	23.05	22.10	2.71	0.596
RBC, 10^12^/L	5–7	6.27	6.57	6.55	0.14	0.246
HGB, g/L	90–130	101.67	109.33	105.50	3.20	0.250
HCT, %	36–43	34.78	37.89	36.77	1.09	0.121
MCV, fL	52–62	56.53	57.67	56.18	0.89	0.499
MCH, pg	17–24	16.52	16.63	16.10	0.31	0.471
MCHC, g/L	290–340	292.17	288.50	286.83	2.04	0.172
PLT, 10^9^/L	200–500	416.00	618.50	522.83	73.74	0.152
RDW-CV, %	-	18.58	19.03	18.57	0.40	0.677

^1^ PQQ·Na_2_: pyrroloquinoline quinone disodium. ^2^ Values are means and pooled SEM (standard error of the mean), *n* = 6. Means within a row without common superscripts differ significantly (*p* < 0.05). ^3^ WBC: white blood cells, RBC: red blood cells, HGB: hemoglobin, HCT: hematocrit, MCV: mean corpuscular volume, MCH: mean corpuscular hemoglobin, MCHC: mean corpuscular hemoglobin concentration, PLT: platelet count, RDW-CV: red blood cell distribution width-coefficient of variance. ^4^ Reference value: from Merck Veterinary Manual.

**Table 5 animals-11-00359-t005:** Effects of graded levels of PQQ·Na_2_
^1^ on serum biochemical parameters of weaned piglets ^2.^

Item ^3^	Reference Value ^4^	PQQ·Na_2_, mg/kg Diet			SEM	*p*-Value
		0	7.5	75.0		ANOVA
day 14						
ALT, U/L	22–47	44.73	48.36	43.30	1.75	0.102
AST, U/L	15–55	58.45	56.08	49.57	3.61	0.204
ALP, U/L	41–176	344.67	404.75	356.78	42.73	0.604
GLU, mmol/L	3.7–6.4	3.07	3.33	4.11	0.38	0.132
TBILI, μmol/L	0.3–8.2	7.49	9.13	9.26	0.92	0.335
TP, g/L	58–83	47.42	47.45	49.10	1.08	0.480
ALB, g/L	23–40	25.87	28.13	27.58	1.02	0.279
UN, mmol/L	2.9–8.8	4.00	3.23	3.40	0.56	0.627
CRE, μmol/L	70–208	71.10	76.03	67.30	4.64	0.436
day 28						
ALT, U/L	22–47	38.75	44.65	42.90	3.22	0.439
AST, U/L	15~55	57.35	57.17	53.52	3.45	0.702
ALP, U/L	41–176	231.38	274.60	243.22	25.69	0.498
GLU, mmol/L	3.7–6.4	4.19	4.79	5.11	0.61	0.580
TBILI, μmol/L	0.3–8.2	3.50	2.85	3.27	0.39	0.514
TP, g/L	58–83	50.73	55.08	57.25	2.28	0.117
ALB, g/L	23–40	27.02	28.55	28.38	1.23	0.652
UN, mmol/L	2.9–8.8	2.95	2.01	2.63	0.50	0.429
CRE, μmol/L	70–208	81.20	83.08	77.37	4.55	0.690

^1^ PQQ·Na_2_: pyrroloquinoline quinone disodium. ^2^ Values are means and pooled SEM (standard error of the mean), *n* = 6. Means within a row without common superscripts differ significantly (*p* < 0.05). ^3^ ALT: alanine aminotransferase, AST: aspartate aminotransferase, ALP: alkaline phosphatase, GLU: glucose, TBILI: total bilirubin, TP: total protein, ALB: albumin, UN: urea nitrogen, CRE: creatinine. ^4^ Reference value: from Merck Veterinary Manual.

**Table 6 animals-11-00359-t006:** Effects of graded levels of PQQ·Na_2_^1^ on organ index of weaned piglets (g/kg BW) ^2.^

Item	PQQ·Na_2_, mg/kg Diet			SEM	*p*-Value
	0	7.5	75.0		ANOVA
Heart	5.87	5.36	5.19	0.25	0.207
Liver	30.81	31.96	28.65	0.97	0.915
Spleen	2.64	2.67	2.52	0.17	0.753
Lung	12.25	11.33	11.72	0.49	0.501
Kidney	5.03	5.44	5.24	0.17	0.234

^1^ PQQ·Na_2_: pyrroloquinoline quinone disodium. ^2^ Values are means and pooled SEM (standard error of the mean), *n* = 6. Means within a row without common superscripts differ significantly (*p* < 0.05).

**Table 7 animals-11-00359-t007:** Antioxidant indexes in serum of weaned pigs fed graded levels of PQQ·Na_2_
^1^ for 28 d ^2^.

Item	PQQ·Na_2_, mg/kg Diet			SEM	*p*-Value
	0	7.5	75.0		ANOVA
day 14					
T-SOD ^3^, U/mL	135.46	142.00	136.23	2.90	0.228
GSH-Px ^4^, U/mL	605.88 ^b^	668.38 ^ab^	695.59 ^a^	25.30	0.025
CAT ^5^, U/mL	2.88 ^b^	4.45 ^a^	3.85 ^ab^	0.45	0.034
T-AOC ^6^, U/mL	5.71 ^b^	7.26 ^a^	7.40 ^a^	0.53	0.038
MDA ^7^, nmol/mL	5.56 ^a^	3.80 ^b^	4.49 ^ab^	0.40	<0.01
day 28					
T-SOD, U/mL	125.31 ^b^	136.23 ^a^	137.23 ^a^	3.00	<0.01
GSH-Px, U/mL	781.99 ^b^	848.90 ^a^	851.84 ^a^	17.21	<0.01
CAT, U/mL	3.17 ^b^	4.19 ^a^	3.59 ^ab^	0.26	0.010
T-AOC, U/mL	5.34 ^b^	6.19 ^ab^	7.24 ^a^	0.47	<0.01
MDA, nmol/mL	5.53 ^a^	3.95 ^b^	3.68 ^b^	0.46	<0.01

^1^ PQQ·Na_2_: pyrroloquinoline quinone disodium. ^2^ Values are means and pooled SEM (standard error of the mean), *n* = 6. Means within a row without common superscripts differ significantly (*p* < 0.05). ^3^ T-SOD: total superoxide dismutase. ^4^ GSH-Px: glutathione peroxidase. ^5^ CAT: catalase. ^6^ T-AOC: total antioxidant capacity. ^7^ MDA: malondialdehyde.

## Data Availability

Some or all data, models, or code generated or used during the study are available in a repository or online in accordance with funder data retention policies (Provide full citations that include URLs or DOIs.).

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
