# Peer review of "Effects of Diet Supplemented with Excess Pyrroloquinoline Quinone Disodium on Growth Performance, Blood Parameters and Redox Status in Weaned Pigs"

_animals, 2021, doi:10.3390/ani11020359_

Round 1
Reviewer 1 Report
Dear Authors:
A very straight forward research project and data outcome. The only comments I have are minor adjustments in sentences (ie writing)
Line 159 .... are presented in Table 4 and 5 and not were present in...
Line 162 is should be are!
Line 275 ....antinutritional factors. (Period here), then Furthermore (new sentence).
Line 286 ... remove the before fedd utilization (not needed)
Line 287 write...........health still needed further research
Line 299 Not a research; just say Research......,
Line 312 end sentence at similar. Then write ...Probably... and then a comma after microorganism then And in lower case....
Author Response
Thanks for your careful reading and thoughtful comments on our previous draft. We have considerate the constructive comments and have revised our manuscript accordingly. Revisions are highlighted in RED in the text.Please see the attachment.

Reviewer 2 Report
Line 27-28: F:G it is feed conversion and not feed efficiency, correct here and in all the manuscript;
Line 28: what is organ index, how you measure general health?
Line 37-42: In both levels of supplementation?
Line 70-71: Improve
Line 85: What was considered for the formation of blocks?
Line 86: Barrows
Line 136: Stats must be better explained, did you evaluated normal distribution? all data was normal?
Line 139: Diarrhea can´t be evaluated by ANOVA.
DIscussion must be improved it is very similar to the introduction, you must discuss your results and not bring a review
Line 289-290: is not correct;
Line 149: pigs receiving the diet with 75 mg of PQQ were not different from the control! (same letters for the stats) so you CAN´T say it was better
Line 151-153: you can´t compare! It was not statistical different.
Table 2 You must present the pig weights;
Table 3 Greater values must be with letter a and lower values with letter b;
Line 173-179: describe better as you have a lot of "ab" saying that is different "b" or from "a" and this is not correct!
Author Response

(The authors gave the same response as above.)

Reviewer 3 Report
This publication is interesting and may be published in the journal ANIMALS, but with significant changes and additions.
General remarks.
Reviewer deficiency refers to the lack of positive control, where the standard animal growth promoter (AGP) would be included in the diet. Change the statistical model and complete the results for male and female in tables 2-7.
Detailed comments:
Line 14 - nutrition is also an environmental factor, too – rewrite,
Line 17 - PQQ · Na2 - provide the full description,
Line 19-20 - a high oral dose - what is the recommended dose?
Line 26 - without any growth promoter? Added
Line 84 - Specify the male and female component
Line 85 - How were the piglets allocated from the sows to the experimental groups?
Line 87 - This is the negative control (without AGP added), where is the positive control (with standard AGP added)? These are the recommended procedures for this type of research. Please explain that!
Line 88/89 - On what basis were such PQQ Na2 additives introduced?
Line 90 – rewrite: exceed NRC [22] recommendations for weaned pigs (Tab. 1).
Line 114 - Specify protection of samples for further analysis.
Line 137 - Improve statistical calculations (analysis). A two-way analysis should be considered: PQQ Na2 addition and gender. The study included gilts (female) and barrows (castrated male). This has not been taken into account in the current statistical model. Please also included gender in tables 2-7. This will be valuable information regarding the gender response to the supplement being tested.
Table 1. Please provide the content of EM in MJ. Lysine total or digestible (available)?
In Table 2-7, include the values for male and female.
In the Discussion and Conclusions, please refer to the results of the influence of PQQ Na2 on the analysed parameters depending on gender.
Please revise the citation of some references from the 1970s - 1990s, e.g. 1,2, 3, 8, 28, 30, 40, et al. Are they really necessary and do they contribute relevant information to the current knowledge?
Author Response

(The authors gave the same response as above.)

Reviewer 4 Report
This manuscript is interesting because explore the effects of PQQ·Na2 on growth performance, blood parameters and redox status in weaned pigs. However it is necessary to indicate why study the effects of this Pyrroloquinoline quinone in piglets.
Simple summary
Line 17: Please include pyrroloquinoline qui-none disodium (PQQ·Na2).
Introduction
Line 54: “PQQ plays an important mission in a lot of physiological and biological functions”. Please, indicate some physiological and biological functions?
Lines 74-76: Please formulate the hypothesis of this study.
Material and Methods
Line 86: Six replicates are low, which compromise the power analysis.
Line 140-141: I understand why the experimental unit is the pen for growth performance parameters and is the animal for the others parameters.
Discussion
Line 273-274: The sentence “Weaning is a vital process for weaned pigs since piglets are exposed to nutritional, psychologic and environmental stresses.” is the same in the summary, lines 13-14. Please correct.
Line 290: The authors did not say that 7.5 and 75 mg PQQ·Na2 improve F:G, because it is not statically different. That is a tendency, maybe.
Line 325: “fortissimo”??
Conclusions
Conclusion needs more improved.
Author Response

(The authors gave the same response as above.)

Reviewer 5 Report
Interesting work, however, begs some questions:
- Have all the pigs completed the experience? Were there losses of piglets?
- Did diarrhea spontaneously pass? Have they been treated?
- What were the causes of diarrhea? After all, they were common in all groups of pigs.
- The ALP value in all groups exceeded the reference values, what could this mean?
- Were piglets weighed individually or were all pigs in the pen weighed together?
- Whether the supplement 75mg PQQ∙Na2/kg feed was not too small for the results of the study, because it is only 118.7mg PQQ∙Na2/kg weight gain, while the Authors cite the dose used in studies in rats: 500-1000mg/kg body weight for females and 1000-2000mg/kg body weight for males.
Author Response

(The authors gave the same response as above.)

Round 2
Reviewer 2 Report
All the suggestions were attended.
Author Response
Thanks for your comments. We have carefully considered your comments and revised the manuscript.
Reviewer 3 Report
General remarks.
In the discussion it should be stated (explained) why there was no positive control group?
Line 54 Anti-oxidative a – better - anti-oxidative a ...,
Line 86-87 - It is not reported how piglets were allocated from the individual sows to the treatment groups, including pen ones.
Line 90 - On what basis were such PQQ Na2 additives introduced? Not explained
Line 138 - Improve statistical calculations (analysis). A two-way analysis should be considered: PQQ Na2 addition and gender. The study included gilts (female) and barrows (castrated male). This has not been taken into account in the current statistical model. Please also included gender in tables 2-7. This will be valuable information regarding the gender response to the supplement being tested. Why was this not done. Please explain or recalculate the available gender values.
Line 152 And There was … - better - And there was…
Table 1. The values in Table 1 were not corrected. What's with lysine? Total or available? Provide the energy value in MJ EM (metabolizable energy).
In Table 2-7, Sex values (gilts and barrows) not provided, why? No source data?
In the Discussion and Conclusions, please refer to the results of the influence of PQQ Na2 on the analysed parameters depending on gender. Not included.

Author Response
Thanks for your comments. We have carefully considered your comments and revised the manuscript.
General remarks.
Point 1: In the discussion it should be stated (explained) why there was no positive control group?
Response 1: Revised accordingly (Line 283-285).
Point 2: Line 54 Anti-oxidative a – better - anti-oxidative a ...,
Response 2: Revised accordingly (Line 54).
Point 3: Line 86-87 - It is not reported how piglets were allocated from the individual sows to the treatment groups, including pen ones.
Response 3: The piglets completely random design in line with body weights and gender on the day of weaning immediately (15 litters of piglets were selected, barrows and gilts were arranged according to their weight. And we selected 54 barrows and 54 gilts from these piglets). Each one of 3 treatments in this study consisted of 6 replicate pens with 6 pigs per pen. The proportion of barrows to gilts was equal in each pen.
Point 4: Line 90 - On what basis were such PQQ Na2 additives introduced? Not explained
Response 4: PQQ·Na2 was premixed with corn starch to a concentration of 1 g/kg mixture before inclusion in the diet. This description in the manuscript Line 93-94.
Point 5: Line 138 - Improve statistical calculations (analysis). A two-way analysis should be considered: PQQ Na2 addition and gender. The study included gilts (female) and barrows (castrated male). This has not been taken into account in the current statistical model. Please also included gender in tables 2-7. This will be valuable information regarding the gender response to the supplement being tested. Why was this not done. Please explain or recalculate the available gender values.
Response 5: In this study, we consider gender were random effects. At the end of this trial, 18 pigs near the average group body weight in each pen were selected randomly to collect sample. It is a great pity that we did not take gender into account in our random selection. However, we will consider the gender component in future studies.
Point 6: Line 152 And There was … - better - And there was…
Response 6: Revised accordingly (Line 152).
Point 7: Table 1. The values in Table 1 were not corrected. What's with lysine? Total or available? Provide the energy value in MJ EM (metabolizable energy).
Response 7: Revised accordingly (Table 1). And lysine data in Table 1 is analyzed values. It’s total lysine.
Point 8: In Table 2-7, Sex values (gilts and barrows) not provided, why? No source data?
Response 8: At the end of this trial, 18 pigs near the average group body weight in each pen were selected randomly to collect sample. It is a great pity that we did not take gender into account in our random selection.
Point 9: In the Discussion and Conclusions, please refer to the results of the influence of PQQ Na2 on the analysed parameters depending on gender. Not included.
Response 9: As above, we had explained in response 5&8. We very thank you for your carefully reading and thoughtful comments. We will consider the gender component in future studies.